# Optimal Arterial Blood Gas Tensions for the Prognosis of Favorable Neurological Outcomes in Survivors after Extracorporeal Cardiopulmonary Resuscitation

**DOI:** 10.3390/jcm11144211

**Published:** 2022-07-20

**Authors:** SungMin Hong, Ji Hoon Jang, Jeong Hoon Yang, Yang Hyun Cho, Joonghyun Ahn, Jeong-Am Ryu

**Affiliations:** 1Division of Pulmonary and Critical Care Medicine, Department of Internal Medicine, Busan Paik Hospital, Inje University College of Medicine, Busan 47392, Korea; hsmsn500@naver.com; 2Division of Pulmonology and Critical Care, Department of Internal Medicine, Inje University Haeundae Paik Hospital, Busan 48108, Korea; saturn80396@gmail.com; 3Department of Critical Care Medicine, Samsung Medical Center, Sungkyunkwan University School of Medicine, Seoul 06351, Korea; jhysmc@gmail.com; 4Division of Cardiology, Department of Medicine, Samsung Medical Center, Sungkyunkwan University School of Medicine, Seoul 06351, Korea; 5Department of Thoracic and Cardiovascular Surgery, Samsung Medical Center, Sungkyunkwan University School of Medicine, Seoul 06351, Korea; yanghyun.cho@samsung.com; 6Biomedical Statistics Center, Data Science Research Institute, Samsung Medical Center, Seoul 06351, Korea; jhguy.ahn@samsung.com; 7Department of Neurosurgery, Samsung Medical Center, Sungkyunkwan University School of Medicine, Seoul 06351, Korea

**Keywords:** arterial blood gas tensions, extracorporeal cardiopulmonary resuscitation, neurological prognosis

## Abstract

Our aim is to assess the optimal levels of oxygen and carbon dioxide for the prognosis of favorable neurologic outcomes in survivors after extracorporeal cardiopulmonary resuscitation (ECPR). We obtained the mean levels of PaCO_2_ and PaO_2_ in arterial blood gas samples 72 h after ECPR. The primary outcome was the neurological status, according to the Cerebral Performance Categories (CPC) scale, upon discharge. Of 119 (48.6%) survivors, 95 (38.8%) had favorable neurologic outcomes (CPC 1 or 2). There was a U-shaped relationship between mean arterial blood gas tensions and poor neurological outcomes. The risk of poor neurological outcome was lowest in patients with the second tertile of mean PaCO_2_ (30–42 mm Hg) and PaO_2_ (120–160 mm Hg). In a multivariable analysis, third tertile of mean PaCO_2_, third tertile of mean PaO_2_, age, shockable rhythm, out of hospital cardiac arrest, duration of cardiopulmonary resuscitation, and ECPR at cardiac catheterization lab were found to be significantly associated with poor neurologic outcomes. Additionally, hypercapnia and extreme hyperoxia were found to be significantly associated with poor neurological outcomes after ECPR. Therefore, maintaining adequate arterial levels of oxygen and carbon dioxide may be important for favorable neurological prognoses in survivors after ECPR.

## 1. Introduction

Neurological prognoses in survivors after successful cardiopulmonary resuscitation (CPR) from cardiac arrest are of the utmost importance [1]. To improve these prognoses, there are several recommendations in the current guidelines for post-cardiac arrest care, including targeted temperature management, control of blood pressure and glucose level, seizure management, oxygenation, and ventilation [2,3]. The guidelines recommend titrating inspired oxygen and ventilation to achieve normal oxygen and carbon dioxide levels after the return of spontaneous circulation [2,3,4,5]. 

Aberrant oxygen levels may be associated with poor prognosis after cardiac arrest [6,7]. Hyperoxia can exacerbate oxygen free radical formation and cause subsequent reperfusion injury [8]. Hypoxia has also been reported to be associated with hospital mortality in survivors after cardiac arrest [6]. Hypercapnia can provoke cerebral vasodilatation and increased cerebral blood flow (CBF), while hypocapnia can lead to cerebral vasoconstriction and decreased CBF [9]. In survivors of cardiac arrest, neurological recovery is related to rapid improvement of CBF to meet the metabolic needs of the brain [10]. Therefore, it is hypothesized that oxygen and carbon dioxide levels may be closely connected with clinical outcomes in survivors after cardiac arrest. 

It remains unknown whether the optimal target of arterial blood gas tension after extracorporeal cardiopulmonary resuscitation (ECPR) is similar to that of conventional CPR. In survivors after ECPR, brain recovery may be related to CBF autoregulation, native circulatory restoration, and amount of extracorporeal membrane oxygenation (ECMO) support [1,11]. However, it is not known in detail how the continuous flow of ECMO affects the autoregulation of CBF [11]. To date, no studies have determined the optimal levels of oxygen and carbon dioxide considering the changes in CBF autoregulation by ECMO. Therefore, in the present study, our aim is to assess the optimal levels of oxygen and carbon dioxide for favorable neurologic outcomes after ECPR.

## 2. Materials and Methods

### 2.1. Study Design and Variables

The present research comprised a retrospective, single-center, and observational study. In this study, ECPR was defined as successful implantation of veno-arterial ECMO and pump-on with cardiac compression during index procedure in patients with cardiac arrest [1,11]. ECPR was considered with confirmed witness arrest, reversible causes of cardiac arrest, and chest compression of 10 min or more [12]. ECPR was contraindicated with the patients with life expectancy less than 6 months, end-stage malignancies, an unwitnessed arrest, or chest compressions of 1 hour or more, but advanced age alone was not an absolute contraindication [11,12]. ECPR was performed when the indications for ECPR were appropriate, regardless of in-hospital or out-of-hospital cardiac arrest [11]. The CPR duration was defined as the total time from onset to halt of chest compression. Targeted temperature management was designated by an on-site intensivist according to the protocol [13].

### 2.2. Study Population

Patients were 18 years or older and underwent ECPR during hospitalization between January 2010 and December 2018 in Samsung Medical Center. All consecutive patients who underwent ECPR during the study period were included in the study. Of these patients, those aged below 18 years, those with Glasgow Coma Scale (GCS) > 12 on admission to the intensive care unit, those with an inappropriate indication for ECPR, those who had pre-existing severe neurologic disease or damage before arrest including traumatic brain injury, major stroke, malignant brain tumor or severe dementia, and those with insufficient medical records were excluded from the study (Figure 1). This study was approved by the Institutional Review Board of Samsung Medical Center (approved No. 2019-10-119). The requirement for informed consent was waived by the Institutional Review Board of Samsung Medical Center due to the retrospective nature of the study.

### 2.3. Data Collection Process

An expert nurse in the ECMO team was responsible for data collection and all data for all patients receiving ECPR were systematically recorded. We obtained the levels of partial pressure of carbon dioxide (PaCO_2_) and partial pressure of oxygen (PaO_2_) in arterial blood gas samples 72 h after ECPR. The sampling interval was determined by the treating physician. The mean PaCO_2_ and PaO_2_ were calculated using entire blood gas measurements 72 h after ECPR. The primary outcome was neurological status upon discharge, as assessed by the Glasgow-Pittsburgh Cerebral Performance Categories (CPC) scale (range: 1 to 5) [1]. 

Favorable neurological outcomes were defined as CPC scores 1 and 2, whereas poor neurologic outcomes were defined as CPC scores of 3 to 5. Two independent intensivists (SMH and JAR) assessed the CPC score by thoroughly reviewing the patient’s medical records. If the investigated scales did not match with each other, the intensivists discussed and corrected the scale.

### 2.4. Statistical Analyses 

Continuous variables are presented as means ± standard deviations, and categorical variables are represented as numbers with subsequent percentages. Data comparisons were carried out using Student’s *t*-test for continuous variables, whereas the Chi-square test was used for categorical variables. We classified the subjects into three groups based on the distribution of the concentration of oxygen and carbon dioxide, using cut-off values between the groups. All possible ranges of oxygen and carbon dioxide were grid-searched to find the cutoffs of both variables. However, since the number of subjects with a low level of carbon dioxide was not sufficient, the cutoff value was fixed at 30 mm Hg, i.e., the generally accepted level [14,15,16,17]. Among all the logistic regression models that included the generated combination of oxygen and carbon dioxide as independent variables, the combination with the largest c-index was selected. For all analyses, multiple logistic regression was performed to correct clinically relevant variables. Eventually, the clinically relevant variables, including mean arterial blood gas tensions, age, first monitored rhythm, type of cardiac arrest, CPR duration, targeted temperature management, and location of ECMO insertion were obtained. All tests were two-sided and *p* values of less than 0.05 were considered statistically significant. Statistical analyses were performed with R Statistical Software (version 4.2.0; R Foundation for Statistical Computing, Vienna, Austria).

## 3. Results

### 3.1. Baseline Characteristics and Clinical Outcomes 

In this study, 245 patients were analyzed (Figure 1). The mean age of the patients was 58.8 ± 15.7 years, and 176 patients (71.8%) were men. Hypertension (48.2%) and diabetes mellitus (35.1%) were identified as the most common comorbidities. Forty patients (16.3%) experienced out-of-hospital cardiac arrest. The mean CPR duration was 26.7 ± 20.6 min. The baseline characteristics of the ECPR patients are presented in Table 1. Compared with the favorable neurologic outcome group, the poor neurologic outcome group comprised elderly patients, higher incidence of out-of-hospital cardiac arrest, initial asystole rhythm and renal replacement therapy, lower incidence of ischemic heart disease as a cause of cardiac arrest, ECPR at cardiac catheterization lab, and longer CPR duration.

Among 245 patients, 119 (48.6%) survived until discharge from the hospital. Of these, 95 (CPC 1 or 2) had favorable neurologic outcomes while 24 (CPC 3 or 4) had poor neurologic outcomes. The entire distribution of CPC scales is shown in Figure 1.

### 3.2. The Relationship between Mean Arterial Blood Gas Tensions and Neurologic Outcomes

Among the three groups of mean PaCO_2_ and PaO_2_, the second tertile groups had the highest number of patients with favorable outcomes (Table 2). There was a U-shaped relationship between the mean arterial blood gas tensions and poor neurological outcomes. The risk of poor neurological outcome was lowest in patients with 30–42 mm Hg of mean PaCO_2_ or 120–160 mm Hg of mean PaO_2_ (Figure 2). Considering mean PaCO_2_ and PaO_2_ simultaneously, the risk of poor neurological outcomes was lowest in patients with the second tertile of mean PaCO_2_ and PaO_2_ (Figure 3). In multivariable analysis, third tertile of mean PaCO_2_ (adjusted odds ratio [OR]: 12.02, 95% confidence interval [CI]: 1.703–84.760), third tertile of mean PaO_2_ (adjusted OR: 2.85, 95% CI: 1.043–7.804), age (adjusted OR: 1.05, 95% CI: 1.021–1.076), shockable rhythm (adjusted OR: 0.16, 95% CI: 0.047–0.555), out of hospital cardiac arrest (adjusted OR: 3.13, 95% CI: 1.074–9.133), CPR duration (adjusted OR: 3.57, 95% CI: 2.263–5.616), and ECPR at cardiac catheterization lab (adjusted OR: 0.17, 95% CI: 0.069–0.409) were found to be significantly associated with poor neurologic outcomes (Table 3).

## 4. Discussion

In the present study, we investigated the optimal levels of oxygen and carbon dioxide for the prognosis of neurologic outcomes after ECPR. The major findings were as follows: First, the risk of poor neurological outcomes was lowest in the second tertile groups: 30–42 mm Hg of mean PaCO_2_ and 120–160 mm Hg of mean PaO_2_. Second, in our multivariable analysis, the third tertile of mean PaCO_2_, the third tertile of mean PaO_2_, age, shockable rhythm, out-of-hospital cardiac arrest, CPR duration, and ECPR at cardiac catheterization lab were found to be significantly associated with poor neurologic outcomes. Finally, there was a U-shaped relationship between the mean arterial blood gas tensions and poor neurological outcomes, but low to normal levels of PaCO_2_ or PaO_2_ demonstrated no independent association with poor neurological outcomes. In contrast, hypercapnia and extreme hyperoxia were found to be independently associated with poor neurological outcomes after ECPR. 

Aberrant arterial blood gas tensions have been reported to be associated with poor neurological outcomes after cardiac arrest, and the impact probably depends on the extent and duration of exposure [7]. Hypocapnia has been demonstrated to be independently associated with poor clinical outcomes [7,15]. Hypocapnia can lead to vasoconstriction, decreased CBF, and cerebral ischemia in neurocritically ill patients [18]. In addition, hypoxia has been reported to be independently associated with hospital mortality after cardiac arrest [6]. In the present study, we observed a U-shaped relationship between mean arterial blood gas tensions and poor neurological outcomes, but hypocapnia and low to normal levels of PaO_2_ demonstrated no independent association with poor neurological outcomes after ECPR. The ECPR survivors with simultaneous low levels of PaCO_2_ and PaO_2_ had favorable neurological outcomes. They had a low incidence of chronic renal disease and high incidences of in-hospital cardiac arrest and ECPR at the cardiac catheterization lab compared with other patients. 

Extreme hyperoxia was found to be independently associated with poor neurological outcomes after ECPR in the present study. Survivors after ECPR may experience severe hyperoxia because oxygenated blood can be easily obtained through the oxygenator of ECMO [19]. Hyperoxia leads to the generation of noxious oxygen radicals and cerebral vasoconstriction, and this mechanism has been reported to provoke brain ischemia [18,20,21]. Early hyperoxia can lead to severe vascular failure, refractory circulatory shock, and exacerbating reperfusion injury after ECPR [22,23,24]. In addition, it is possible to obtain fully oxygenated blood through an oxygenator at the right radial arterial line if the survivor’s native cardiac function is severely depressed after ECPR. Therefore, hyperoxia may reflect markedly impaired cardiac function in such patients. It has been proposed that hyperoxia may be associated with poor neurological outcomes in survivors after ECPR [22,23,24]. 

A few studies have reported that hypercapnia may be associated with a greater likelihood of favorable outcomes [7,25]. However, hypercapnia may lead to cerebral vasodilatation and increased CBF, and might provoke cerebral edema in patients with impaired CBF autoregulation [9]. Another study reported that hypercapnia was common after cardiac arrest and demonstrated an independent association with poor neurological outcomes after resuscitation from cardiac arrest [15]. Therefore, further studies are needed to demonstrate whether hypercapnia during postcardiac arrest care improves the clinical outcome [14]. 

In our previous study, old age (>65 years) and prolonged CPR duration (>30 min) were related to poor neurologic outcomes after ECPR [1]. There was a significant interaction between age and CPR duration in predicting neurological outcomes after ECPR [1]. Despite initial hypoxic-ischemic injury, the considerable reserve and tolerance of young brains to hypoxic-ischemic injury may lead to favorable neurological outcomes [26]. In addition, age-induced altered cerebral hemodynamics may affect neurologic recovery after cardiac arrest [1]. In the present study, ECMO insertion in cardiac catheterization lab was associated with favorable neurologic outcomes. This may be associated with short ECMO pump-on time. A catheterization lab is the best place for ECPR in a short time; ECMO insertion can be safe and highly effective in this setting [27,28]. In our previous studies, we found that initial arrest rhythm and low-flow time may be associated with neurological outcome after ECPR [1,29]. Notably, shockable rhythm was associated with favorable neurological outcomes after ECPR [1,29]. ECPR patients after in-hospital cardiac arrest may have more favorable outcomes than those after out-of-hospital cardiac arrest [30]. Patients may have more prolonged arrest to ECMO pump-on time after out-of-hospital cardiac arrest compared with after in-hospital cardiac arrest.

This study has several limitations. First, it was conducted over a long period. In the meantime, post-cardiac arrest management may have evolved, which may have affected the clinical outcomes during the study period. Second, the effects of continuous ECMO flow on cerebral autoregulation were unclear in this study. Lastly, it is hypothesized that extreme hyperoxia may be obtained in ECPR patients with a markedly impaired heart. Extremely high oxygen levels in the right radial artery may mainly be caused by the flow of ECMO. It is unclear whether the poor clinical outcomes were related to hyperoxia itself or impaired heart function. Although the present study provides valuable insights, prospective large-scale studies are needed to evaluate the optimal levels of oxygen and carbon dioxide for favorable neurologic outcomes after ECPR.

## 5. Conclusions

In this study, we observed a U-shaped relationship between poor neurological outcomes and mean arterial blood gas tensions in the first 72 h after ECPR. However, hypocapnia, normocapnia, hypoxia, and normoxia demonstrated no relationship with poor neurological outcomes, whereas hypercapnia and extreme hyperoxia were found to be significantly associated with poor neurological outcomes. Therefore, it is proposed that maintaining adequate arterial levels of oxygen and carbon dioxide may be important for favorable prognoses of neurological outcomes in survivors who underwent ECPR. 

## Figures and Tables

**Figure 1 jcm-11-04211-f001:**
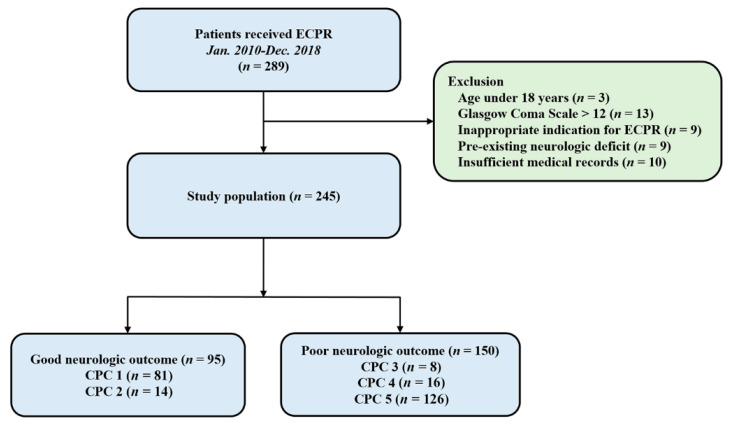
Study flow chart. ECPR, extracorporeal cardiopulmonary resuscitation; CPC, Cerebral Performance Categories scale.

**Figure 2 jcm-11-04211-f002:**
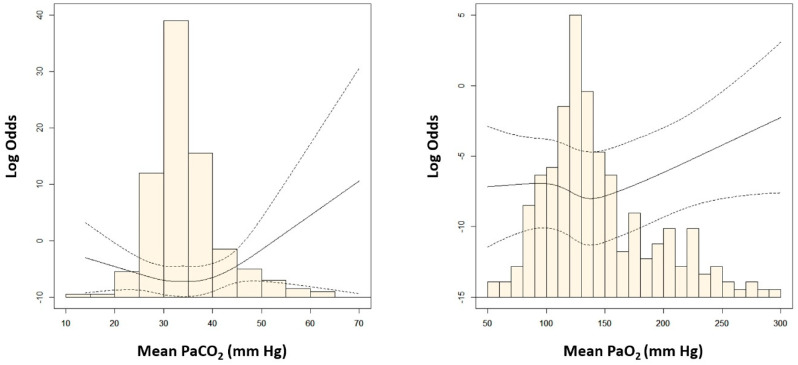
Adjusted odds ratio (OR) of poor neurologic outcomes according to the mean levels of arterial carbon dioxide (PaCO_2_) and arterial oxygen (PaO_2_). OR was log−transformed to reduce skewness.

**Figure 3 jcm-11-04211-f003:**
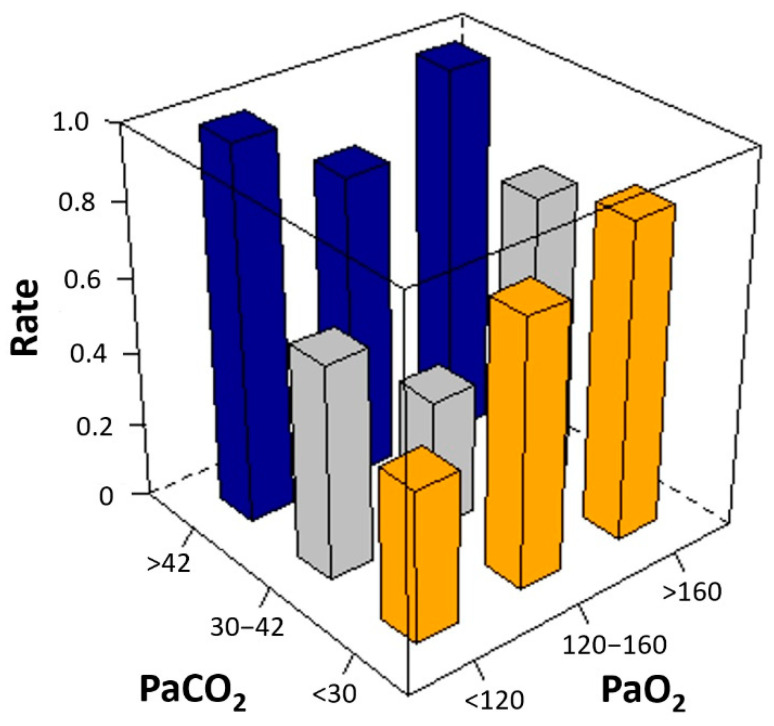
Unadjusted rate of poor neurological outcomes according to the mean levels of arterial carbon dioxide (PaCO_2_) and arterial oxygen (PaO_2_). Considering mean PaCO_2_ and PaO_2_ simultaneously, the risk of poor neurological outcomes was lowest in patients with the second tertile of mean PaCO_2_ and PaO_2_ (the smallest gray square column in the middle).

**Table 1 jcm-11-04211-t001:** Baseline characteristics of patients.

	Favorable Neurological Outcome (*n* = 95)	Poor Neurological Outcome (*n* = 150)	*p*-Value
Age (years)	56.6 ± 15.4	60.1 ± 15.8	0.089
Gender, male	73 (76.8)	103 (68.7)	0.215
Comorbidities			
Hypertension	45 (47.4)	73 (48.7)	0.947
Diabetes mellitus	32 (33.7)	54 (36.0)	0.816
Previous myocardial infarction	22 (23.2)	34 (22.7)	0.999
Current smoker	21 (22.1)	24 (16.0)	0.302
Malignancy	13 (13.7)	30 (20.0)	0.274
Chronic kidney disease ^a^	8 (8.4)	26 (17.3)	0.076
Dyslipidemia	15 (15.8)	17 (11.3)	0.416
CPR details			
Type of cardiac arrest			0.033
Out of hospital cardiac arrest	9 (9.5)	31 (20.7)	
In-hospital cardiac arrest	86 (90.5)	119 (79.3)	
First monitored rhythm			0.006
Asystole	6 (6.3)	30 (20.0)	
Pulseless electrical activity	47 (49.5)	74 (49.3)	
Shockable rhythm (VT or VF)	42 (44.2)	46 (30.7)	
CPR duration (min)	16.63 ± 14.42	33.13 ± 21.37	<0.001
Cardiac cause of arrest			0.015
Ischemic	54 (56.8)	58 (38.7)	
Non-ischemic	17 (17.9)	31 (20.7)	
Management in the intensive care unit			
Targeted temperature management	18 (18.9)	30 (20.0)	0.970
Arctic Sun	10 (10.5)	24 (16.0)	
Cooling pad	8 (8.4)	6 (4.0)	
Intra-aortic balloon pump	10 (10.5)	8 (5.3)	0.205
Renal replacement therapy	26 (27.4)	83 (55.3)	<0.001
Location of ECPR			<0.001
Cardiac catheterization lab	39 (41.1)	24 (16.0)	
Intensive care unit	29 (30.5)	74 (49.3)	
Emergency department	22 (23.2)	45 (30.0)	
Others (operation room, general wards, etc.)	5 (5.3)	7 (4.7)	

^a^ Chronic kidney disease is defined as either kidney damage or glomerular filtration rate less than 60 mL/min/1.73 m^2^ for 3 months or longer. Reported are *n* (%) for categorical variables and mean ± standard deviation for continuous variables. CPR: cardiopulmonary resuscitation; VT: ventricular tachycardia; VF: ventricular fibrillation; ECPR: extracorporeal cardiopulmonary resuscitation.

**Table 2 jcm-11-04211-t002:** Comparison of mean blood gas tension distribution between the favorable neurological outcome group and poor neurological outcome group.

Mean Blood Gas Tension (mm Hg)	Favorable Neurological Outcome (*n* = 95)	Poor Neurological Outcome (*n* = 150)	*p*-Value
PaCO_2_	33.19 ± 4.06	36.71 ± 14.41	0.005
Tertile of mean PaCO_2_			<0.001
First tertile (<30)	18 (18.9)	37 (24.7)	
Second tertile (30–42)	75 (78.9)	84 (56.0)	
Third tertile (>42)	2 (2.1)	29 (19.3)	
PaO_2_	131.68 ± 30.91	148.40 ± 55.19	0.003
Tertile of mean PaO_2_			<0.001
First tertile (<120)	32 (33.7)	49 (32.7)	
Second tertile (120–160)	52 (54.7)	47 (31.3)	
Third tertile (>160)	11 (11.6)	54 (36.0)	

Reported are *n* (%) for categorical variables and mean ± standard deviation for continuous variables. PaCO_2_: the mean level of arterial carbon dioxide 72 h after cardiac arrest; PaO_2_: the mean level of arterial oxygen 72 h after cardiac arrest.

**Table 3 jcm-11-04211-t003:** Multivariable analysis of factors associated with poor neurological outcomes.

	Adjusted OR (95% CI)	*p*-Value
Tertile of mean PaCO_2_		
First tertile (<30 mm Hg)	1	Reference
Second tertile (30–42 mm Hg)	0.51 (0.205–1.283)	0.148
Third tertile (>42 mm Hg)	12.02 (1.703–84.760)	0.012
Tertile of mean PaO_2_		
First tertile (<120 mm Hg)	1	Reference
Second tertile (120–160 mm Hg)	0.50 (0.223–1.107)	0.083
Third tertile (>160 mm Hg)	2.85 (1.043–7.804)	0.039
Age (years)	1.05 (1.021–1.076)	<0.001
First monitored rhythm		
Asystole	1	Reference
Pulseless electrical activity	0.31 (0.092–1.056)	0.058
Shockable rhythm (VT or VF)	0.16 (0.047–0.555)	0.003
Out of hospital cardiac arrest	3.13 (1.074–9.133)	0.034
CPR duration ^a^	3.57 (2.263–5.616)	<0.001
ECMO insertion in cardiac catheterization lab	0.17 (0.069–0.409)	<0.001

^a^ CPR duration was log−transformed to reduce skewness. OR: odds ratio; CI: confidence interval; PaCO_2_: mean level of arterial carbon dioxide 72 h after ECPR; PaO_2_: mean level of arterial oxygen 72 h after ECPR; VT: ventricular tachycardia; VF: ventricular fibrillation; CPR: cardiopulmonary resuscitation; ECMO: extracorporeal membrane oxygenation.

## Data Availability

Regarding data availability, our data are available on the Harvard Dataverse Network (http://dx.doi.org/10.7910/DVN/P7FF28) as recommended repositories (The dataset was published at 20 June 2022).

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
