# Peer review of "Optimal Arterial Blood Gas Tensions for the Prognosis of Favorable Neurological Outcomes in Survivors after Extracorporeal Cardiopulmonary Resuscitation"

_jcm, 2022, doi:10.3390/jcm11144211_

Round 1

Reviewer 1 Report

Aim

Evaluating is not an objective but a method. the authors must use the appropriate terminology and the applicability of the study

material and methods must be modified: 

study design: is indicated in study population. It must appear in the first section.

Ethical approval. is indicated in study population.  It must appear in the last section.

Definitions and outcomes: You must differentiate between definitions (they must be described in the introduction) and variables (in methods). the authors describe definitions in methods.

Process: authors must include a section describing the process

Discussion

future proposals

Too generic: (further studies are needed). develop more; types of studies or variables neededsome of the limitations described are not limitations or are not justified:

"This was a retrospective study. Therefore, CPC was measured retrospectively between two investigators based on medical records." a retrospective study is not a limitation. in the case of not being able to recover all the data that would be a limitation

single institution. not a limitation

bibliography

As far as possible, authors should include actualized bibliography

Author Response

July 18 2022

Ms.Alina Leng, Section Managing Editor

Journal of Clinical Medicine

Manuscript ID: jcm-1809347

Title: Optimal arterial blood gas tensions for the prognosis of favorable neurological outcomes in survivors after extracorporeal cardiopulmonary resuscitation

Dear Ms.Alina Leng

Thank you very much for your letter and for the helpful comment from the reviewer. We appreciate the opportunity to resubmit our revised manuscript entitled “Optimal arterial blood gas tensions for the prognosis of favorable neurological outcomes in survivors after extracorporeal cardiopulmonary resuscitation”. As always, you and your editorial staff have again provided us with a comprehensive and prompt review. Many of the valuable and constructive points that the reviewers pointed out were well taken by all the authors. After going over the reviewer’s comments, my colleagues and I have performed additional investigation and made some revisions in hopes of improving our paper. The revised and added portions of the manuscript are stated in the “Response to Reviewers” and are underlined and highlighted in the revised manuscript for your convenience. In addition, we have revised the manuscript based on the iThenticate report you sent us.

All authors contributed to the conception and interpretation of data, drafting of the manuscript, revising it critically for important intellectual content, and final approval of the manuscript. The whole manuscript or part of it, neither has been published and is not being considered for publication elsewhere in any language except as an abstract. None of the authors have any financial relationships with any company or any other bias or conflict of interest.

We believe that these findings have scientific and clinical impact and will be interesting and informative to your readers. We hope that, upon review, our study will be found to be meritorious of publication in the Journal of Clinical Medicine.

Yours sincerely,

Jeong-Am Ryu, M.D., Ph.D.

Department of Critical Care Medicine and Department of Neurosurgery, Samsung Medical Center, Sungkyunkwan University School of Medicine, 81 Irwon-ro, Gangnam-gu, Seoul 06351, Republic of Korea

Tel: 82-2-3410-6399, Fax: 82-2-2148-7088

E-mail: lamyud.ryu@samsung.com

Response to Reviewers

Reviewer #1:

Aim

Evaluating is not an objective but a method. the authors must use the appropriate terminology and the applicability of the study

  1. We apologize for the use of inappropriate terminology. As your recommendation, we revise the clear statement in Abstract and Introduction. We revised manuscript and added the following sentences in the Discussion section of the revised manuscript (line 37-39 in page 3 and line 79-80 in page 4-5).

Our aim is to assess the optimal levels of oxygen and carbon dioxide for prognosis of the favorable neurologic outcomes in survivors after extracorporeal cardiopulmonary resuscitation (ECPR).

Therefore, in the present study, our aim is to assess the optimal levels of oxygen and carbon dioxide for favorable neurologic outcomes after ECPR.

material and methods must be modified:

study design: is indicated in study population. It must appear in the first section.

Ethical approval. is indicated in study population.  It must appear in the last section.

Definitions and outcomes: You must differentiate between definitions (they must be described in the introduction) and variables (in methods). the authors describe definitions in methods.

Process: authors must include a section describing the process

  1. We agree with the reviewer’s comment. We have revised the order of each paragraph and content. We revised the section of Materials and Methods as following sentences (line 82-127 in page 5-7).

  1. Materials and Methods

2.1. Study Design and variables

This study was a retrospective, single-center, and observational study. In this study, ECPR was defined as successful implantation of veno-arterial ECMO and pump-on with cardiac compression during index procedure in patients with cardiac arrest. ECPR was considered with confirmed witness arrest, reversible causes of cardiac arrest, and chest compression of 10 minutes or more. ECPR was contraindicated with the patients with life expectancy less than 6 months, end-stage malignancies, an unwitnessed arrest, chest compressions of 1 hour or more, but advanced age alone was not absolute contraindication of ECPR. ECPR was performed when the indications for ECPR were appropriate, regardless of in-hospital or out-of-hospital cardiac arrest. The CPR duration was defined as the total time from onset to halt of chest compression. The targeted temperature management was determined by an on-site intensivist according to the protocol.

2.2. Study Population

Patients were 18 years or older who underwent ECPR during hospitalization between January 2010 and December 2018 in Samsung Medical Center. All consecutive patients who underwent ECPR during the study period were included in the study. Of these patients, patients aged below 18 years, those with Glasgow Coma Scale (GCS) > 12 on admission to intensive care unit, those with an inappropriate indication for ECPR, those who had pre-existing severe neurologic disease or damage before arrest including traumatic brain injury, major stroke, malignant brain tumor or severe dementia, and those with insufficient medical records were excluded from the study (Figure 1). This study was approved by the Institutional Review Board of Samsung Medical Center (approved No. 2019-10-119). The requirement for informed consent was waived by the IRB of Samsung Medical Center due to the retrospective nature of the study.

2.3. Data Collection process

An expert nurse in ECMO team was responsible for data collection and all data for all patients receiving ECPR were systematically recorded. We obtained the levels of partial pressure of carbon dioxide (PaCO2) and partial pressure of oxygen (PaO2) in arterial blood gas samples during 72 hr after ECPR. The sampling interval was based on the discretion of the treating physician. The mean PaCO2 and the mean PaO2 were calculated using the entire blood gas measurements during 72 hr after ECPR. The primary outcome was neurological status upon discharge, as assessed by the Glasgow-Pittsburgh Cerebral Performance Categories (CPC) scale (range: 1 to 5).

Favorable neurological outcomes were defined as CPC scores 1 and 2, whereas poor neurologic outcomes were defined as CPC scores of 3 to 5. Two independent intensivists (SMH and JAR) assessed the CPC score by thoroughly reviewing the patient’s medical records. If the investigated scale did not match with each other, the intensivists discussed and corrected the scale.

Discussion

future proposals

Too generic: (further studies are needed). develop more; types of studies or variables needed some of the limitations described are not limitations or are not justified:

"This was a retrospective study. Therefore, CPC was measured retrospectively between two investigators based on medical records." a retrospective study is not a limitation. in the case of not being able to recover all the data that would be a limitation

single institution. not a limitation

  1. We agree with the reviewer’s comment. In this study, two independent intensivists (SMH and JAR) assessed the CPC score by thoroughly reviewing the patient’s medical records. If the investigated scale did not match with each other, the intensivists discussed and corrected the scale. We agree with the reviewer’s comment about single institution and retrospective nature. As your recommendation, we have removed unnecessary content. We revised the limitation section as the following sentences (line 275-284 in page 15).

This study has several limitations. First, this study was conducted over a long period. In the meantime, post-cardiac arrest management may have been more advanced than before, which may have affected the clinical outcomes during the study period. Second, the effects of continuous ECMO flow on cerebral autoregulation were unclear in this study. Lastly, it is hypothesized that extreme hyperoxia may be obtained in ECPR patients with a markedly impaired heart. Extremely high oxygen levels in the right radial artery may mainly be caused by the flow of ECMO. It is unclear whether the poor clinical outcomes are related to hyperoxia itself or impaired heart function. Although it still provides valuable insight, prospective large-scale studies are needed to evaluate optimal levels of oxygen and carbon dioxide for favorable neurologic outcomes after ECPR.

bibliography

As far as possible, authors should include actualized bibliography

  1. As your recommendation, we revised the references. Particularly, the references that were too outdated data have been removed. We also used available references as far as possible.

We thank the reviewer for valuable comments. Addressing them fully has significantly strengthened the manuscript.  

Reviewer 2 Report

Dear authors, I read your article with interest. First of all, this well-designed study will contribute to the literature. However, I do have a few criticisms. Although the main subject of this study is the effect of gas retention on neurological complications, there are other parameters associated with poor neurological outcomes, as seen in the multivariant analysis. For example age, place of ecpr or duration of cpr. Each of these parameters deserves a separate study. However, despite the fact that you have these data in your discussion,you did not mention these issues at all. We would like to see information in the discussion section about these factors, especially those associated with poor neurological outcomes.

Author Response

July 18 2022

Ms.Alina Leng, Section Managing Editor

Journal of Clinical Medicine

Manuscript ID: jcm-1809347

Title: Optimal arterial blood gas tensions for the prognosis of favorable neurological outcomes in survivors after extracorporeal cardiopulmonary resuscitation

Dear Ms.Alina Leng

Thank you very much for your letter and for the helpful comment from the reviewer. We appreciate the opportunity to resubmit our revised manuscript entitled “Optimal arterial blood gas tensions for the prognosis of favorable neurological outcomes in survivors after extracorporeal cardiopulmonary resuscitation”. As always, you and your editorial staff have again provided us with a comprehensive and prompt review. Many of the valuable and constructive points that the reviewers pointed out were well taken by all the authors. After going over the reviewer’s comments, my colleagues and I have performed additional investigation and made some revisions in hopes of improving our paper. The revised and added portions of the manuscript are stated in the “Response to Reviewers” and are underlined and highlighted in the revised manuscript for your convenience. In addition, we have revised the manuscript based on the iThenticate report you sent us.

All authors contributed to the conception and interpretation of data, drafting of the manuscript, revising it critically for important intellectual content, and final approval of the manuscript. The whole manuscript or part of it, neither has been published and is not being considered for publication elsewhere in any language except as an abstract. None of the authors have any financial relationships with any company or any other bias or conflict of interest.

We believe that these findings have scientific and clinical impact and will be interesting and informative to your readers. We hope that, upon review, our study will be found to be meritorious of publication in the Journal of Clinical Medicine.

Yours sincerely,

Jeong-Am Ryu, M.D., Ph.D.

Department of Critical Care Medicine and Department of Neurosurgery, Samsung Medical Center, Sungkyunkwan University School of Medicine, 81 Irwon-ro, Gangnam-gu, Seoul 06351, Republic of Korea

Tel: 82-2-3410-6399, Fax: 82-2-2148-7088

E-mail: lamyud.ryu@samsung.com

Response to Reviewers

Reviewer #2:

Dear authors, I read your article with interest. First of all, this well-designed study will contribute to the literature. However, I do have a few criticisms. Although the main subject of this study is the effect of gas retention on neurological complications, there are other parameters associated with poor neurological outcomes, as seen in the multivariant analysis. For example age, place of ecpr or duration of cpr. Each of these parameters deserves a separate study. However, despite the fact that you have these data in your discussion, you did not mention these issues at all. We would like to see information in the discussion section about these factors, especially those associated with poor neurological outcomes.

  1. First, we thank the reviewer for valuable comments. In this study, third tertile of mean PaCO2, third tertile of mean PaO2, age, shockable rhythm, out of hospital cardiac arrest, duration of cardiopulmonary resuscitation, and ECPR at cardiac catheterization lab were identified to be significantly associated with poor neurologic outcomes in multivariable analysis. However, except arterial blood gas tensions, we did not explain why these variables were associated with poor neurological outcomes. We add the following sentences at the Discussion section in revised manuscript (line 259-274 in page 14-15). Although our report still provides valuable insight, prospective large-scale studies are needed to evaluate optimal blood gas tensions and clinical variables for favorable neurologic outcomes after ECPR.

In our previous study, old age (> 65 years) and prolonged CPR duration (>30 min) were related to poor neurologic outcomes after ECPR [1]. There was a significant interaction between age and CPR duration in predicting neurological outcomes after ECPR [1]. Despite initial hypoxic-ischemic injury, the considerable reserve and tolerance of young brains to hypoxic-ischemic injury may lead to favorable neurological outcomes [15]. In addition, age-induced altered cerebral hemodynamics may affect neurologic recovery after cardiac arrest [1]. In present study, ECMO insertion in cardiac catheterization lab was associated with favorable neurologic outcomes. It may be associated with short ECMO pump-on time. A catheterization lab is the best place for ECPR in a short time and ECMO insertion can be safe and highly effective in this place [26,27]. In our previous studies, initial arrest rhythm and low-flow time may be associated with neurological outcome after ECPR [1,28]. Especially, shockable rhythm was associated with favorable neurological outcomes after ECPR [1,28]. ECPR patients after in-hospital cardiac arrest may have more favorable outcomes than those after out-of-hospital cardiac arrest [29]. The patients after out-of-hospital cardiac arrest may have more prolonged arrest to ECMO pump-on time compared with those after in-hospital cardiac arrest.

We thank the reviewer for valuable recommendation. Addressing them fully has significantly strengthened the manuscript.
